# Safety assessment of a novel C-type natriuretic peptide derivative and the mechanism of bone- and cartilage-specific toxicity

**Takafumi Yotsumoto**[1,2]*, **Naomi Morozumi**[1], **Ryuichi Nakamura**[1,2], **Toshimasa Jindo**[1,2], **Mayumi Furuya**[1], **Yasuyuki Abe**[1,2], **Tomonari Nishimura**[1,2], **Hiroaki Maeda**[1,2], **Hiroyuki Ogasawara**[1,2], **Yoshiharu Minamitake**[1], **Kenji Kangawa**[3]

1 Asubio Pharma Co., Ltd., Kobe, Japan, 2 Daiichi Sankyo Co., Ltd., Tokyo, Japan, 3 National Cerebral and Cardiovascular Center Research Institute, Osaka, Japan

* yotsumoto.takafumi.ur@daiichisankyo.co.jp

**Data Availability Statement:** All relevant data are within the manuscript and its Supporting Information files.

## Abstract

ASB20123, a C-type natriuretic peptide/ghrelin chimeric peptide, was designed as a novel peptide and demonstrated full agonistic activity for natriuretic-peptide receptor B and a significantly longer half-life in plasma compared with the native peptide. We researched the toxicological profile of ASB20123, the correlation between the morphological change of the epiphyseal plate and bone and cartilage toxicity, and biomarkers to detect the toxicity. ASB20123 was systemically administered to male and female rats at daily dose levels of 0.5, 1.5, and 5.0 mg/kg/day for 4 weeks. In this study, toxicity was observed as changes related to bone and cartilage tissues, and no other toxicological changes were observed in all animals. Next, ASB20123 was administered to 12-month-old rats with a little epiphyseal plate. The toxic changes related to bone and cartilage tissues were not observed in any animal with a closed epiphyseal plate, indicating that the toxic changes were triggered by the growth-accelerating effect on the bone and cartilage. Furthermore, we searched for the biomarker related to the bone and cartilage toxicity using rats treated with ASB20123 at doses of 0.005, 0.05, 0.5, and 5.0 mg/kg/day for 4 weeks. A close correlation between necrosis/fibrosis in the epiphysis and metaphysis and thickness of the epiphyseal plate in the femur was confirmed in this study. A decrease in the bone mineral density (BMD) of the femur also was associated with the appearance of bone toxicity. These results indicated that the toxicity of ASB20123 was limited to bone- and cartilage-specific changes, and these changes were triggered by an excessive growth accelerating effect. Furthermore, our data suggested that the thickness of the epiphyseal plate and BMD could be reliable biomarkers to predict bone toxicity.

## Introduction

The C-type natriuretic peptide (CNP) analog is one of the most exciting therapeutic approaches to treat achondroplasia [1]. The binding of CNP to natriuretic-peptide receptor B

**Funding:** The authors received no specific funding for this work. Asubio pharma Co., Ltd and/or Daiichi Sankyo Co., Ltd provided support in the form of salaries for authors [TY, MN, RN, TJ, MF, YA, TN, HM, HO, YM], but did not have any additional role in the study design, data collection and analysis, decision to publish, or preparation of the manuscript. The specific roles of these authors are articulated in the 'author contributions' section.

**Competing interests:** KK has nothing to disclose. TY, MN, RN, TJ, MF, YA, TN, HM, HO and YM are employed by Asubio pharma Co., Ltd and/or Daiichi Sankyo Co., Ltd. There are no patents, products in development or marketed products to declare. This does not alter our adherence to PLOS ONE policies on sharing data and materials.

(NPR-B) inhibits fibroblast growth factor receptor 3 downstream signaling [2], recognized as an important regulator of endochondral bone growth [3]. We recently reported that the exogenous administration of CNP-53 has the potential to stimulate skeletal growth related to short stature, and restore the skull morphology and size of foramen magnum in CNP-KO rats [4, 5]. As a CNP derivative, a clinical trial utilizing BMN-111 is currently proceeding in pediatric patients with achondroplasia [6–8].

We designed ASB20123, a CNP/ghrelin chimeric peptide, as a novel peptide. ASB20123 contains the full-length 22-amino acids of human CNP-22 fused to the 17-amino acids on the C-terminus region of human ghrelin, and the single amino acid is substituted in its ghrelin region. The application of C-terminal part of ghrelin resulted in the higher stability of CNP analogs, compared to that of CNP-22 as the native form; it also improved their bioactivity as stimulators of endochondral bone growth [9–11]. Furthermore, the C-terminal part of the ghrelin includes a BX7B motif, the cluster of basic amino acids might be implicated in the interaction with hyaluronic acid molecules which are the components of extracellular matrix in growth plate [12]. ASB20123, this novel CNP derivative demonstrated full agonistic activity for NPR-B and showed significantly longer half-life in plasma compared with the native forms. A significant and dose-dependent increase in body length was shown in rats after 12 weeks of administration via subcutaneous infusion [13].

CNP is produced in the brain, kidney, bone, blood cells, blood vessels, and heart [14]. NPR-B is expressed in the brain, lung, bone, heart, and ovary. It is also expressed at relatively high levels in fibroblasts and vascular smooth muscle cells [15]. However, the changes that occur after excessive exogenous CNP exposure remain to be clarified. Furthermore, the toxicological profile of the CNP derivative has not been reported previously. In the present study, we evaluated the exhaustive toxicological profile of ASB20123. Furthermore, we researched the relationship between the specific bone and cartilage toxicity and morphological changes of the epiphyseal plate and reliable biomarkers to detect the toxicity.

## Materials and methods

### Test article

ASB20123 (GLSKGCFGLKLDRIGSMSGLGCVQQRKDSKKPPAKLQPR, C6 and C22 are bound as an intramolecular disulfide bond) was produced by Asubio Pharma Co. Ltd., Japan using a recombinant DNA method in *Escherichia coli*, purified with high-performance liquid chromatography, and verified by amino acid composition analysis and amino acid sequence analysis [13]. The purity of ASB20123 was 98.4%. Acetate buffer (0.03 mol/L) was added with 10 w/v% sucrose and 1 w/v% benzyl alcohol and used as a vehicle. All chemicals and regents used in the present study were purchased from FUJIFILM Wako Pure Chemical Industries, Ltd., Japan and Otsuka Pharmaceutical Factory, Inc., Japan.

### Animals

Sprague-Dawley (SD) rats were purchased from Charles River Laboratories Japan, Inc., Japan and were used for the studies conducted at the Nonclinical Research Center, LSI Medience Corporation, Japan and Asubio Pharma Co., Ltd, Japan. The animals were housed individually in stainless-steel cages in a humidity-, temperature- and ventilation-controlled environment with an automatic 12-h light/dark cycle. They were provided with a standard, pelleted lab chow diet (CRF-1, Oriental Yeast Co., Ltd., Japan) and tap water *ad libitum*. Rats were acclimatized for at least 1 week prior to the study. Clinical sign was observed at least once a day during the experiment period by trained animal care staff. Throughout all studies, the following criteria were used to exclude an animal from the study and humanely euthanize to prevent

undue pain or distress: weight loss (>20%) and moribund condition. No animals were excluded by these criteria, but one animal was found dead on Day 7 of administration period. All animals except for one animal found dead were euthanized and necropsied after blood collection at the end of administration or recovery period. The animals were euthanized by exsanguination from the carotid artery under anesthesia with isoflurane (Mylan Inc., USA). All animal experiments were conducted in accordance with the Guidelines for Animal Experiments of LSI Medience Corporation, Japan and/or Asubio Pharma Co., Ltd., Japan and were approved by the Institutional Animal Care and Use Committee of LSI Medience Corporation, Japan and/or the Committees for Ethics in Animal Experiments of Asubio Pharma Co., Ltd., Japan, respectively.

## Study protocol and administration

**Toxicity profiling study (study 1).** Twenty male and 20 female rats at 7 weeks of age were randomly divided into 4 groups each with 5 males and 5 females. Dose levels were set at 0.5, 1.5, and 5.0 mg/kg/day. The control animals were treated with the vehicle at the same dosing volume as the test article-treated animals. The dosing volume was set at 1 mL/kg, and the dosing volume for each animal was calculated based on the most recent body weight. Each animal was injected with the dosing formulation subcutaneously once a day for 4 weeks into the back of the animals using a disposable injection needle and syringe. The dose levels were selected based on the result of previous study, in which the rats received at the dose of ASB20123 0.15 mg/kg/day for 12 weeks showed over-growth [13].

**Mechanism study (study 2).** Fifteen male and 15 female rats at 12 months of age were randomly divided into 3 groups, each with 5 males and 5 females. Dose levels were set at 0.5 and 5.0 mg/kg/day. Other conditions and procedures were the same as those for study 1.

**Biomarker and recovery study (study 3).** Forty male rats at 7 weeks of age were randomly divided into 5 groups with 5 males at the doses of 0, 0.005, 0.05, 0.5, and 5.0 mg/kg/day for biomarker study, and 3 groups with 5 males at the doses of 0, 0.5, and 5.0 mg/kg/day for recovery study withdrawal period for 13 weeks, respectively. Other conditions and procedures were the same as those for study 1.

## Observations and examination items

In study 1, clinical observation, measurements of body weight, food and water consumption, ophthalmology, urinalysis, hematology, blood chemistry, measurements of serum alkaline phosphatase (ALP) isozymes and osteocalcin, body length (naso-anal length), bone mineral density (BMD), organ weight, necropsy, and histopathology analyses were conducted. In studies 2 and 3, clinical observation, measurements of body weight, body length, femur bone length, and BMD, necropsy, and histopathology of the femur and tibia were conducted.

**Clinical observation.** Clinical signs and mortality were observed once or more per day during the administration and recovery period.

**Blood chemistry.** Blood samples were collected from the posterior vena cava and centrifuged at $1870 \times g$ for 10 minutes to obtain serum samples. The total protein, albumin, A/G ratio, total bilirubin, asparate aminotransferase, alanine aminotransferase, gamma glutamyl-transpeptidase, alkaline phosphatase, lactate dehydrogenase, creatine phosphokinase (CPK), total cholesterol, triglycerides, phospholipids, glucose, blood urea nitrogen, creatinine, inorganic phosphorus, and calcium were examined with an auto-analyzer (7170, Hitachi Ltd., Japan), and the sodium, potassium, and chloride were examined with an electrolyte analyzer (EA07, A&T Corporation, Japan).

**ALP isozymes and osteocalcin.** The remaining serum samples collected for blood chemistry were used for the serum ALP isozyme and osteocalcin measurement. For ALP isozyme measurement, the auto electrophoresis system (Epalyzer 2, Helena Laboratories Co., Ltd., USA) was used. For osteocalcin measurement, an immunoradiometric assay was applied with the Rat Osteocalcin IRMA kit (Immutopics, International, LLC., USA).

**Body length (naso-anal length and femur).** At the end of the administration period, naso-anal lengths of each rat were examined with a scale after euthanasia. Femoral lengths were measured with digital calipers after removal at necropsy.

**Bone mineral density (BMD).** Bone mineral density (cortical and sponge) of the isolated left femur of each rat was measured using CT scanning (Latheta LCT-200; Hitachi Aloka Medical, Japan).

**Histological examination.** Whole rat specimens were embedded in paraffin, sectioned, stained with hematoxylin and eosin (HE), and examined microscopically. The thickness of the epiphyseal plate at the proximal end of the femur was measured under a light microscope. It was measured at nine sites for the proximal end of the femur. The average thickness was considered the epiphyseal plate thickness for each rat.

## Statistical analysis

Data were analyzed for homogeneity of variance by Bartlett's test. If the variance was homogeneous, Dunnett's test was performed for multiple comparisons between the control group and each test article group. If the variance was heterogeneous by Bartlett's test, Steel's test was performed for multiple comparisons between the control group and each test article group. A two-tailed test was used as Bartlett's test, and $P$ values less than 0.05 were considered statistically significant.

## Results

### Toxicity profile of ASB20123 in study 1

In the clinical observation, abnormal gait appearing as a shuffling gait of the hind limbs was observed in all the test article-treated groups at the dose level of 0.5 mg/kg/day or more at the 3rd week of administration and later. The number of animals exhibiting these symptoms increased dose-dependently. The BMD values of both cortical and trabecular bone in the femur were significantly low in all the test article-treated groups compared with the control group (Fig 1).

The results of histopathological examination are shown in S1 Table, and representative histopathological findings for the proximal femoral bone in rats are shown in Fig 2. The test article-related changes were observed in the bone/cartilage tissues as follows. In the femur, thickening of the epiphyseal plate was observed at the dose levels of 0.5 mg/kg/day or more. This change involved both the proximal and distal portions of the femur, was intense in the proximal portion, and was concomitant with increases in the primary bone and osteoblasts. In the proximal portion, there was necrosis of the epiphysis/metaphysis, fibrosis in the marrow of the head, and ectopic chondrogenesis/osteogenesis. In the tibia, thickening of the epiphyseal plate and an increase in the primary bone were observed in all the test article-treated groups. These changes involved both the proximal and distal portions of the tibia and were intense or highly frequent in the distal portion. There were also increases in osteoblasts in the proximal and distal portions in males, and necrosis of the epiphysis/metaphysis and inflammatory changes in the surrounding tissues in the distal portion.

The changes in body length, CPK, ALP activity, ALP-isozyme fraction, and osteocalcin values are shown in Fig 3. Significantly high values of body length were shown in males at the

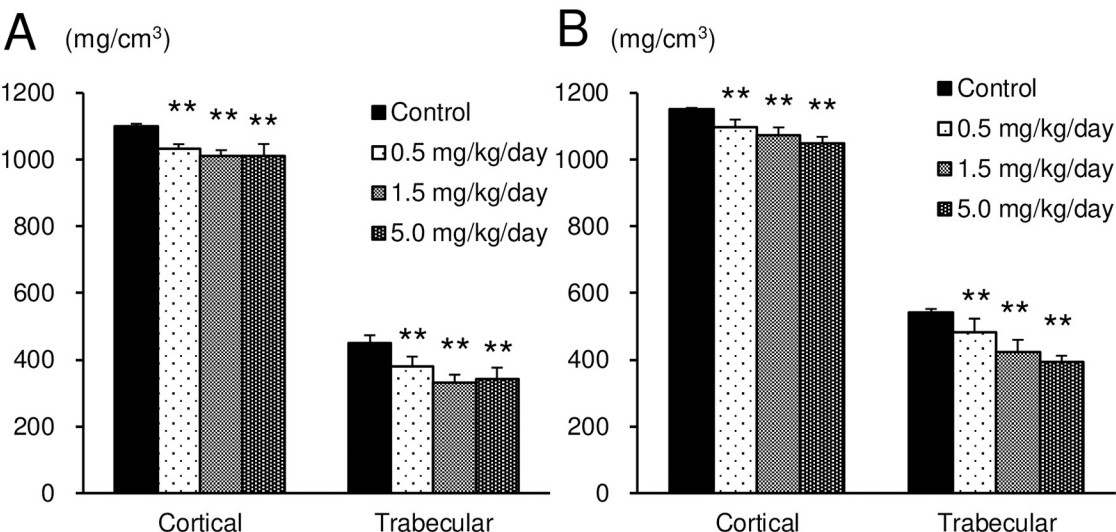

**Fig 1. BMD of both the cortical and trabecular bone in the femurs of male (A) and female (B) rats treated subcutaneously with ASB20123 for 4 weeks in study 1.** Each value represents the mean ± SD of 5 rats, [**] $P < 0.01$ vs. vehicle-treated group by Dunnett's multiple comparison test.

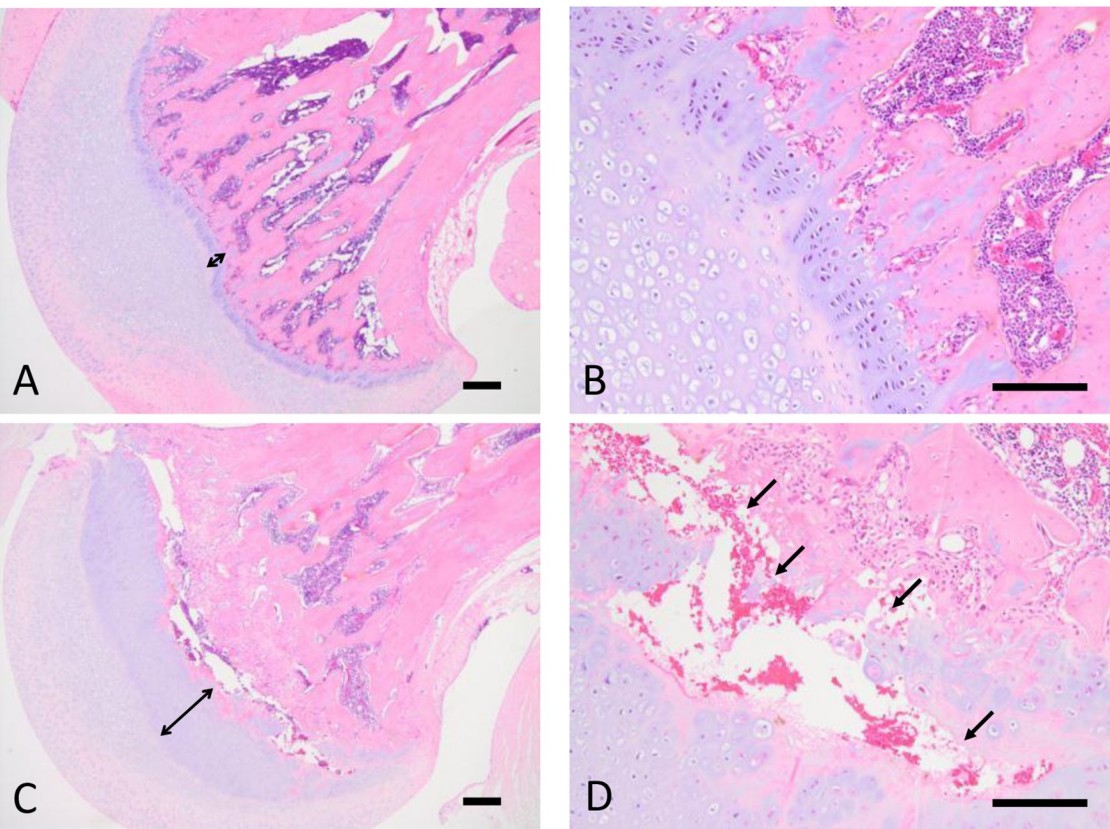

**Fig 2. Representative histopathological findings in the proximal femoral bone in rats in study 1.** (A) Vehicle group (× 40). (B) Vehicle group (× 100). (C) 0.5 mg/kg/day group (× 40). (D) 0.5 mg/kg/day group (× 100). Bidirectional arrows indicate the width of the epiphyseal plate. Arrows indicate the necrosis of cartilage/osseous tissues. Scale bars represent 200 μm.

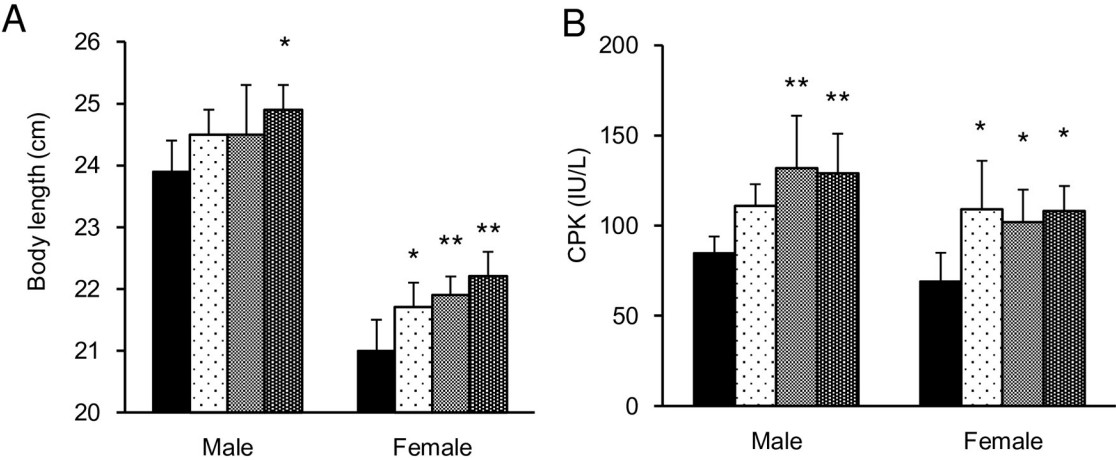

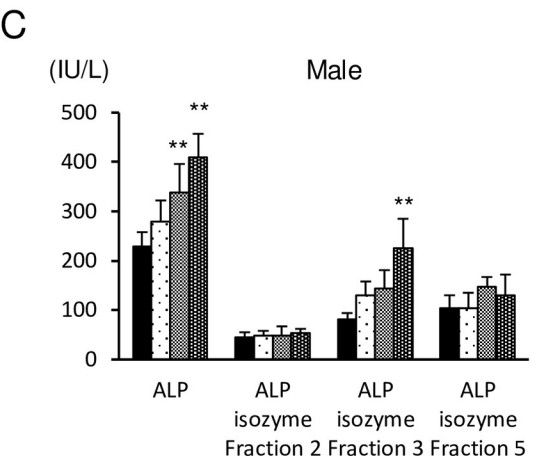

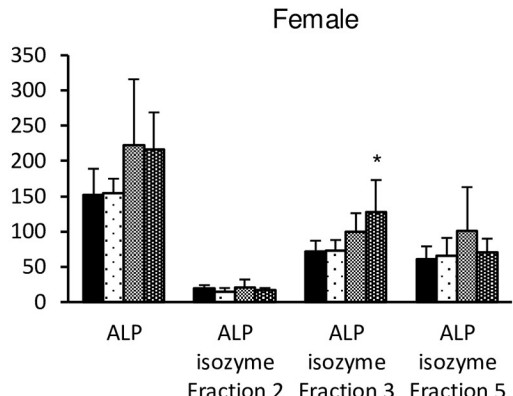

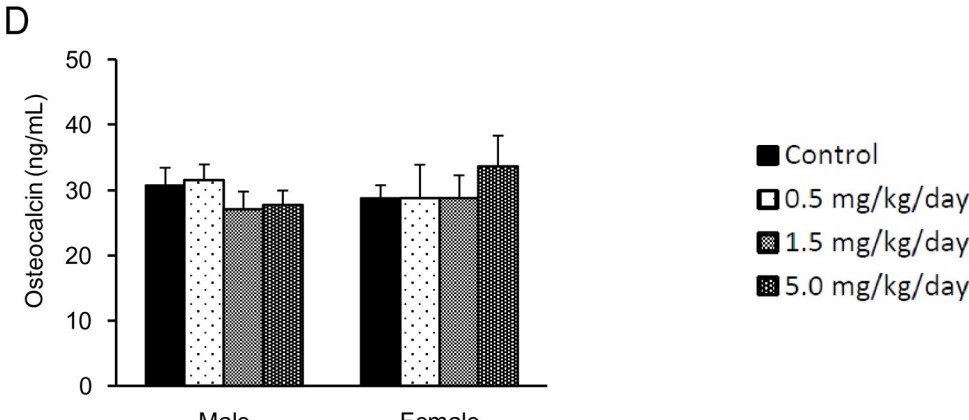

**Fig 3. Effects of ASB20123 on the body length (A), CPK activity (B), ALP and ALP-isozyme fraction activity (C), and osteocalcin value (D) in rats treated subcutaneously for 4 weeks in study 1.** Each value represents the mean ± SD of 5 rats, * $P < 0.05$, ** $P < 0.01$ vs. vehicle-treated group by Dunnett's multiple comparison test.

dose levels of 5.0 mg/kg/day and in females at 0.5 mg/kg/day and more, compared with the vehicle control group. The CPK activity was significantly high or showed a tendency toward higher in all test article treated groups. The ALP activity was significantly high in males at the dose levels of 1.5 and 5.0 mg/kg/day. Although there was no statistically significant difference, the same tendency was observed in males at the dose levels of 0.5 mg/kg/day and in females at 1.5 and 5.0 mg/kg/day. The ALP-isozyme fraction 3 was significantly high in males and females at the dose levels of 5.0 mg/kg/day compared with the vehicle control group, and the same tendency was observed in males administered 0.5 and 1.5 mg/kg/day and females administered 1.5 mg/kg/day. Meanwhile, there were no changes in other items of blood chemistry and the osteocalcin concentrations in any group compared to the control group.

No changes were observed in any group in body weight, food and water consumption, ophthalmology, urinalysis, hematology, necropsy, and organ weight.

## Mechanism of the specific bone and cartilage toxicity in study 2

ASB20123 was administered to the 12-month-old rats. One animal was found dead on Day 7 due to the formation of a tumor in the duodenum. The epiphyseal plate closure was observed in all observation sites of the vehicle group of the aged rats, except for the proximal tibia in 2 female rats. The number of rats with the remaining epiphyseal plate in the treatment group was larger than that in the vehicle group. Increases in osteoblasts and primary bone and degeneration/necrosis in the epiphysis and metaphysis were observed in some animals with the epiphyseal plate, but these findings were not observed in animals with a closed epiphyseal plate (Table 1). No test article-related changes were observed in the clinical observation, body weight, femur bone length, and BMD, but abnormal gait was observed in only 1 animal in the clinical observation in the 0.5 mg/kg/day dosage group.

## Searching for the biomarker related to the bone and cartilage toxicity and evaluating for its recovery in study 3

The thickness of the epiphyseal plate, femur bone length, and bone mineral density of the femur were measured in the rats treated with ASB20123 at doses of 0.005, 0.05, 0.5, and 5.0 mg/kg/day for 4 weeks. The epiphyseal plate thickness increased in a dose-dependent manner, and the toxic findings in the epiphysis and metaphysis were observed only in individuals with thickening of more than 200 μm of the epiphyseal plate. A decrease in the BMD in the femur also reflected the appearance of bone toxicity. In contrast, a correlation between the appearance of bone toxicity and body or femur lengths was not observed (Fig 4).

The recovery of low BMD value in the femur and bone toxicity observed in the rats treated with ASB20123 for 4 weeks were assessed after 13 weeks withdrawal period. The results of clinical observation, measurements of body weight, body length, femur bone length, BMD, necropsy, and histopathology of the femur and tibia after 4 weeks administration period were similar to those of study 1 (Fig 5A and S2 Table). At the end of the recovery period, necrosis of the epiphysis/metaphysis in the proximal portion of femur and deformation in distal tibia end joint were observed in some individuals (1 or 2 animals/groups) of ASB20123 treated groups (S3 Table). The BMD values of cortical bone were increased during the recovery period from 1112.1 mg/cm$^3$ to 1305.1 mg/cm$^3$ in the control group, from 1079.7 mg/cm$^3$ to 1290.6 mg/cm$^3$ in the 0.5 mg/kg/day group, from 1009.6 mg/cm$^3$ to 1282.4 mg/cm$^3$ in the 5 mg/kg/day group, respectively, it was significantly low in the 5.0 mg/kg/day group. The BMD values of trabecular bone were changed during the recovery period from 456.0 mg/cm$^3$ to 481.1 mg/cm$^3$ in the control group, from 397.7 mg/cm$^3$ to 396.0 mg/cm$^3$ in the 0.5 mg/kg/day group, from 324.6

**Table 1. Histopathological findings in the femur and tibia in study 2.**

| Sex | | Male | | | Female | |
|---|---|---|---|---|---|---|
| Group | Vehicle | | ASB20123 | Vehicle | | ASB20123 |
| Organs / Tissues — Dose (mg/kg/day) | 0 | 0.5 | 5.0 | 0 | 0.5 | 5.0 |
| Findings* — No. of animals | 5 | 5 | 4# | 5 | 5 | 5 |
| **Femur (proximal)** | | | | | | |
| Epiphyseal plate closure | 5 | 4 | 4 | 5 | 3 | 0$ |
| Thickening, epiphyseal plate | - | 1 (+) | - | - | 2 (+/++) | 1$ (++) |
| Increase, osteoblast and primary bone | - | - | - | - | 2 (+) | - |
| Degeneration/necrosis, epiphysis/metaphysis | - | - | - | - | 1 (+) | 5 (+/+++) |
| **Femur (distal)** | | | | | | |
| Epiphyseal plate closure | 5 | 5 | 2 | 5 | 2 | 1 |
| Thickening, epiphyseal plate | - | | 2 (+++) | - | 3 (+/++) | 4 (++/+++) |
| Increase, osteoblast and primary bone | - | - | 2 (+) | - | 3 (+) | 4 (+/++) |
| **Tibia (proximal)** | | | | | | |
| Epiphyseal plate closure | 5 | 0 | 0 | 3 | 0 | 0 |
| Thickening, epiphyseal plate | - | 5 (+) | 4 (+) | - | 1 (+) | 5 (+) |
| Increase, osteoblast and primary bone | - | 3 (+) | 4 (+) | - | 5 (+) | 5 (+) |
| **Tibia (distal)** | | | | | | |
| Epiphyseal plate closure | 5 | 5 | 5 | 5 | 5 | 5 |

Grades: -, normal; +, slight; ++, moderate; +++, severe; +/++, slight to moderate; +/+++, slight to severe; ++/+++, moderate to severe. The numbers of animals with histopathological findings are listed. Vehicle: 0.03 mol/L acetic acid buffer solution (pH 4) containing 10 w/v% sucrose and 1 w/v% benzyl alcohol.

*: No test article-related changes were observed in any animal without an epiphyseal plate.

#: One animal was found dead on Day 7 due to the formation of a tumor in the duodenum.

$: The findings of epiphyseal plate were not evaluated in the proximal femoral bone of 4 rats, because the specimen did not have the target tissue.

mg/cm$^3$ to 395.5 mg/cm$^3$ in the 5 mg/kg/day group, respectively, it was significantly low in the 0.5 and 5.0 mg/kg/day group (Fig 5B).

## Discussion

ASB20123 is a CNP derivative, and this peptide stimulates bone growth through proliferation and differentiation of chondrocytes [13]. In this study, ASB20123 was administered to male and female rats at daily dose levels of 0.5, 1.5, and 5.0 mg/kg/day for 4 weeks to investigate its toxicity. In this study, toxic changes were observed in the bone and cartilage tissues, and no other toxic changes were observed in all animals.

In the histopathological examination, thickening of the epiphyseal plate, which was frequently accompanied by increases in the primary bone and osteoblasts, was observed in the femur and tibia in all the test article-treated groups. Similar cartilage thickening was also detected in the temporomandibular joint and sternum. These findings were characteristic, prominent, and therefore considered to be primary changes based on the pharmacological action of ASB20123. In relation to the above osseous changes, the body length was extended in all the test-article treated groups, and serum ALP-isozyme fraction activity increased, since it is derived from bone and is elevated in the serum as a result of various bone diseases and bone growth. In addition to the above changes due to a pharmacological action of ASB20123, the following toxicity findings were observed in this study. In the proximal portion of the femur, there was necrosis of the epiphysis/metaphysis, fibrosis in the marrow of the head, and ectopic chondrogenesis/osteogenesis in all the test article-treated groups. Necrosis of the trabecula/

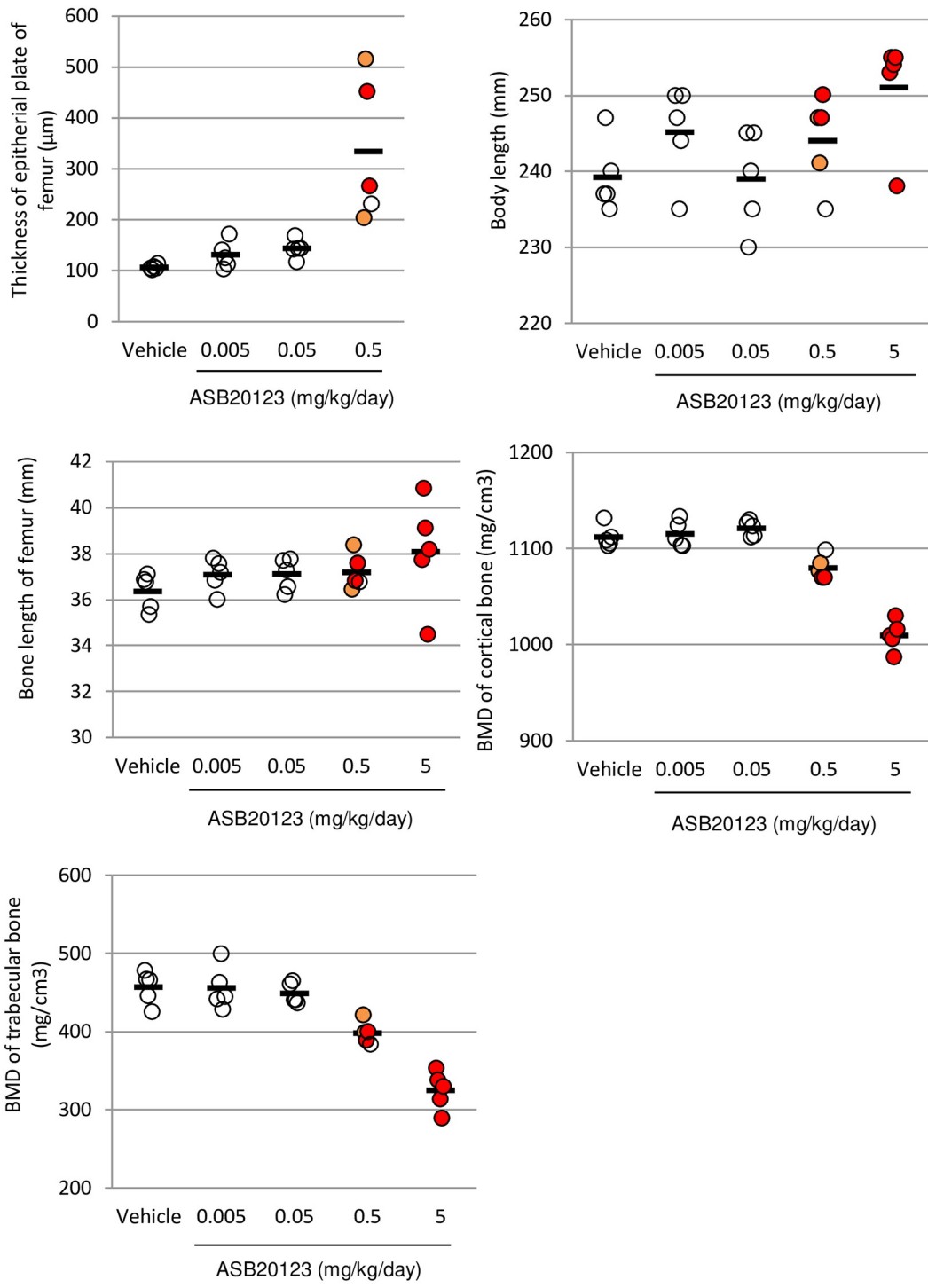

**Fig 4. The correlation between bone and cartilage toxicity and several parameters in study 3.** The thickness of the epiphyseal plate of the femur (A), body length (B), and bone length of the femur (C). The BMD of the cortical bone (D) and trabecular bone (E) in the femurs is shown. Bone toxicity observed in each animal is shown in the colored circle, the open circle represents no toxicity, orange indicates slight toxicity, and red indicates severe toxicity. Each bar represents the mean of 5 rats.

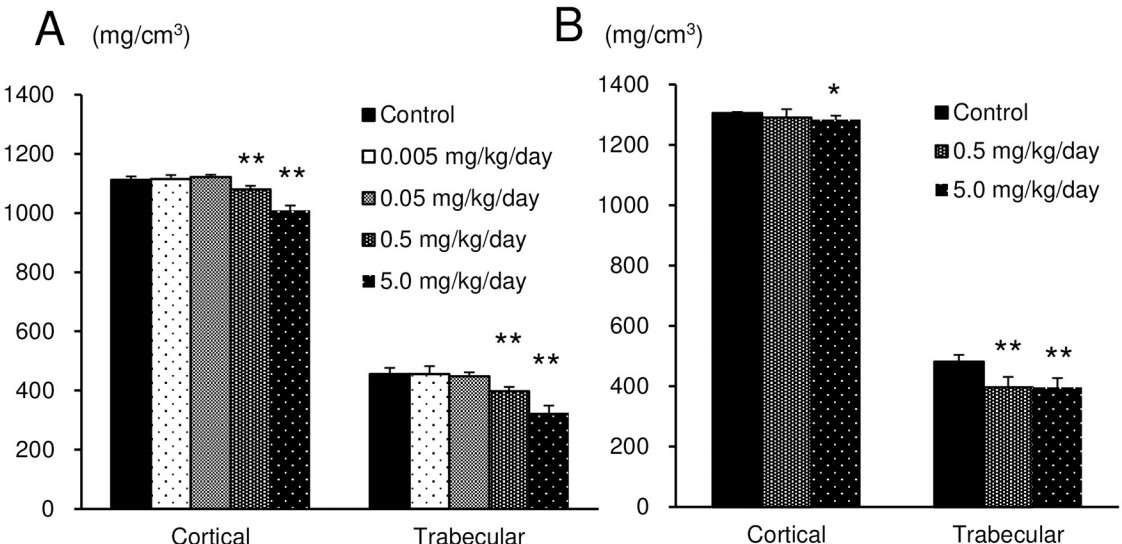

**Fig 5. BMD of both the cortical and trabecular bone in the femurs of rats treated subcutaneously with ASB20123 for 4 weeks followed by 13 weeks recovery period in study 3.** (A) At the end of administration period. (B) At the end of recovery period. Each value represents the mean ± SD of 5 rats, ** $P < 0.01$, * $P < 0.05$ vs. vehicle-treated group by Dunnett's multiple comparison test.

marrow in the epiphysis of the head and degeneration/necrosis of the peripheral muscle fibers was also sporadically observed. In the tibia, there was necrosis of the epiphysis/metaphysis and inflammatory changes in the surrounding tissues in the distal portion. All these findings associated with the above-mentioned primary changes were probably inflammatory, ischemic, or reactive due to the physical or physiological stimulation following thickening of the epiphyseal plate. These changes were intense or localized in the proximal portion, especially in the head of the femur and in the distal portion of the tibia, suggesting a possibility that the landing-shock on the hind limbs during walking/moving accelerates these bone/cartilage changes. Clinical signs revealed shuffling gait of the hind limbs. This symptom was considered to be caused by the excessive cartilage increase in the distal tibia. The increase of CPK activity might be ascribable to degeneration or necrosis of the muscle with rapid extension of various bones; however, the increase of CPK activity was slight and was judged not to be of serious toxicological significance.

To research the involvement of the epiphyseal plate in the bone-related changes, ASB20123 was administered to the 12-month-old rats with a little epiphyseal plate. The epiphyseal plate closure was observed in all examination sites of the vehicle group, except for in the proximal tibias of 2 female rats. The number of rats with the remaining epiphyseal plate in the test article-treated group was larger than that in the vehicle group. It was suggested that the administration of ASB20123 delayed the epiphyseal plate closure, and this result corresponded to those of our previous reports [4, 5]. An increase in osteoblasts and primary bone and degeneration/necrosis in the epiphysis and metaphysis were not observed in any animal with a closed epiphyseal plate. These results indicated that the toxic changes in the bone and cartilage tissues were triggered by the excessive growth-accelerating effect based on the pharmacological action of ASB20123.

Biomarkers related to bone and cartilage toxicity, thickness of the epiphyseal plate, body length, femur bone length, and BMD of the cortical and trabecular bone in the femur were measured in the rats treated with ASB20123 at the doses of 0.005, 0.05, 0.5, and 5.0 mg/kg/day for 4 weeks. A reliable correlation between necrosis/fibrosis in the epiphysis and metaphysis

and thickness of the epiphyseal plate of the femur was confirmed in this study. A decrease in BMD in the cortical bone in the femur was also relevant to the bone and cartilage toxicity. These parameters might be good markers to predict bone- and cartilage-specific toxic changes, because the thickness of the epiphyseal plate can be monitored using radiographic examination, computed tomography, and magnetic resonance imaging in humans [16, 17].

While the decrease of BMD value could be a reliable biomarker to predict bone- and cartilage toxicity, we need to take care of it in the treatment of achondroplasia, because the lower BMD value of achondroplasia population was reported [18]. At the end of the recovery period for 13 weeks, necrosis of the epiphysis/metaphysis in the proximal portion of femur and deformation in distal tibia end joint were still observed in some individuals of ASB20123 treated groups. The incidence of bone toxicity in femur and tibia decreased and considered reversible. Meanwhile, some individuals could not recover; it might be because physical stress exacerbated the lesion through moving freely in the housing environment of stainless-steel cage. The BMD values in the ASB20123 treated groups did not recover completely to the value of control group after 13 weeks recovery period. However, the BMD values in the ASB20123 treated groups increased similarly to those in the control group for 13 weeks. These results indicate that at least a further reduction in BMD could be prevented by discontinuation of treatment.

In this study, we evaluated the toxic profile of ASB20123 the CNP derivative with an extended half-life. As a result, over-dosing of ASB20123 induced excessive growth acceleration through endochondral bone growth, resulting in bone- and cartilage-specific toxicity changes in normal young rats without closed epiphyseal plates. Furthermore, our data suggested that the thickness of the epiphyseal plate and BMD of the cortical bone could be reliable biomarkers to predict bone- and cartilage-specific toxicity. The dosage regimen of CNP derivative would be a key factor for the success as therapeutic drug.

## Supporting information

**S1 Table. Histopathological findings in rats treated subcutaneously with ASB20123 for 4 weeks in study 1.**
(DOC)

**S2 Table. Histopathological findings of femur and tibia in rats treated subcutaneously with ASB20123 for 4 weeks in study 3.**
(DOC)

**S3 Table. Histopathological findings of femur and tibia in rats treated subcutaneously with ASB20123 for 4 weeks followed by 13 weeks recovery period in study 3.**
(DOC)

## Author Contributions

**Conceptualization:** Takafumi Yotsumoto, Naomi Morozumi, Hiroyuki Ogasawara, Kenji Kangawa.

**Data curation:** Takafumi Yotsumoto, Naomi Morozumi, Ryuichi Nakamura, Toshimasa Jindo, Yasuyuki Abe, Tomonari Nishimura, Hiroyuki Ogasawara.

**Formal analysis:** Takafumi Yotsumoto, Ryuichi Nakamura, Toshimasa Jindo, Yasuyuki Abe, Tomonari Nishimura.

**Investigation:** Takafumi Yotsumoto, Naomi Morozumi, Ryuichi Nakamura, Toshimasa Jindo, Yasuyuki Abe, Tomonari Nishimura.

**Methodology:** Takafumi Yotsumoto, Naomi Morozumi, Ryuichi Nakamura, Toshimasa Jindo, Mayumi Furuya, Yasuyuki Abe, Tomonari Nishimura, Hiroyuki Ogasawara.

**Project administration:** Naomi Morozumi, Mayumi Furuya, Hiroaki Maeda, Hiroyuki Ogasawara.

**Resources:** Hiroaki Maeda, Hiroyuki Ogasawara.

**Supervision:** Mayumi Furuya, Hiroaki Maeda, Yoshiharu Minamitake.

**Visualization:** Kenji Kangawa.

**Writing – original draft:** Takafumi Yotsumoto.

**Writing – review & editing:** Naomi Morozumi, Mayumi Furuya, Yasuyuki Abe, Hiroyuki Ogasawara, Kenji Kangawa.

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
