## [Decision Letter · Decision Letter 0]

28 Jun 2019

PONE-D-19-14712

Safety assessment of a novel C-type natriuretic peptide derivative and the mechanism of bone- and cartilage-specific toxicity

PLOS ONE

Dear Mr Yotsumoto,

Thank you for submitting your manuscript to PLOS ONE. After careful consideration, we feel that it has merit but does not fully meet PLOS ONE’s publication criteria as it currently stands. Therefore, we invite you to submit a revised version of the manuscript that addresses the points raised during the review process.

We would appreciate receiving your revised manuscript by Aug 12 2019 11:59PM. To enhance the reproducibility of your results, we recommend that if applicable you deposit your laboratory protocols in protocols.io, where a protocol can be assigned its own identifier (DOI) such that it can be cited independently in the future. For instructions see: http://journals.plos.org/plosone/s/submission-guidelines#loc-laboratory-protocols

We look forward to receiving your revised manuscript.

Kind regards,

Michael Bader

Academic Editor

PLOS ONE

**Journal Requirements:**

2.  At this time, we request that you  please report additional details in your Methods section regarding animal care, as per our editorial guidelines: 1) Please provide details of animal welfare (e.g., shelter, food, water, environmental enrichment) 2) please describe any steps taken to minimize animal suffering and distress, such as by administering analgesics, 3) please include the method of sacrifice and 4) Please describe thecare received by the animals, including the frequency of monitoring and the criteria used to assess animal health and well-being. Thank you for your attention to these requests.

**Comments to the Author**

1. Is the manuscript technically sound, and do the data support the conclusions?

Reviewer #1: Yes

Reviewer #2: Yes

2. Has the statistical analysis been performed appropriately and rigorously? 

Reviewer #1: Yes

Reviewer #2: Yes

3. Have the authors made all data underlying the findings in their manuscript fully available?

Reviewer #1: Yes

Reviewer #2: Yes

4. Is the manuscript presented in an intelligible fashion and written in standard English?

Reviewer #1: Yes

Reviewer #2: Yes

5. Review Comments to the Author

Reviewer #1: Yotsumoto et al provide toxicity data for a novel C-type natriuretic analog that may have utility in treating achondroplasia. Pathological changes observed included necrosis within the metaphysis, fibrosis in the marrow head, and ectopic chondrogenesis/osteogenesis. These pathological events were confined to animals with active growth plates and potentially the result of excessive growth velocity. This is a generally well-written paper that provides new and useful information.

Specific Comments.

1/ Introduction, line 44:- replace “expecting” with “exciting” This line should read “…(CNP) analog is one of the most exciting therapeutic …”.

2/ The reviewer suggest more background information regarding the rational for using a CNP/ghrelin chimeric peptide should be provided eg describe the potential for interaction with hyaluronic acid.

3/ Page 12, line 191:- There is no mention in the results section of any differences in blood chemistry measurements observed between the groups eg triglycerides (mentioned in the methods section). If there were no significant differences, the reviewer suggests line 191 could read “… ophthalmology, blood chemistry, urinalysis …”.

4/ It would be helpful for the reader if the age of the rats was included in the title of Supporting Table 1 or alternatively the title to the table read “Study 1 – histopathology in rats treated subcutaneously with …”.

Reviewer #2: The manuscript titled “Safety assessment of a novel C-type natriuretic peptide derivative and the mechanism of bone and mineral specific toxicity” by Yotsumoto et al describes experiments injecting a new GC-B activator in 12-month old rats and examining effects in bone and cartilage tissue. These data are of potential interest to the field because they suggest that the dosing regimen of the CNP analog may be important for successful clinical results.

Major issues:

Figure 4 of this manuscript shows a very interesting inverse relationship in response to ASB20123 infusion with the drug increasing bone length and decreasing density in both the cortical and trabecular bone. If this is the case, then it may limit the use of the drug because while it increases long bone length, it also results in less bone mineral density, which is unfavorable.

It would be very interesting to see if the BMD increases with time after the initial CNP analog infusion period has ended. In other words, if ASB20123 increases bone length, does it always also decrease bone mineral density or does it return to normal values after the ASB20123 treatment is discontinued. To test this, the investigators should analyze another group of animals 6 weeks after the last ASB20123 injection and determine if BMR is still reduced or whether it increased in the absence of the drug. The question being addressed with this experiment is whether any increase in bone length resulting from ASB20123 infusion also results in permanent reductions in BMD? If the later scenario is observed, then this could limit the utility of ASB20123 in the treatment of achondroplasia if it always results in less dense, and presumably, weaker bone.

Minor issues:

The manuscript would benefit by editing from a native English-speaking person. Trabecular bone is the newer and more common term for “spongy” bone.

6. PLOS authors have the option to publish the peer review history of their article (what does this mean?). If published, this will include your full peer review and any attached files.

Reviewer #1: No

Reviewer #2: No

---

## [Author Response · Author response to Decision Letter 0]

24 Jul 2019

Journal requirement

Response.

According to journal requirement, we have checked our revised manuscript to meet PLOS ONE’s style requirements. 

At this time, we request that you please report additional details in your Methods section regarding animal care, as per our editorial guidelines: 1) Please provide details of animal welfare (e.g., shelter, food, water, environmental enrichment) 2) please describe any steps taken to minimize animal suffering and distress, such as by administering analgesics, 3) please include the method of sacrifice and 4) Please describe the care received by the animals, including the frequency of monitoring and the criteria used to assess animal health and well-being. Thank you for your attention to these requests.

Response.

According to journal requirement, we revised the methods section about animal care. Details of animal welfare (e.g., shelter, food, water, environmental enrichment) were mentioned in line 88-92, the method of sacrifice was in line 92-94, and frequency of monitoring was in line 132-133 in the Revised Manuscript with Track Changes, respectively. The steps taken to minimize animal suffering and distress and the criteria used to assess animal health and wellbeing were described in the Guidelines for Animal Experiments of LSI Medience Corporation and/or Asubio Pharma Co., Ltd, and these studies were conducted in accordance with these guidelines. 

We are grateful to the Reviewer #1 for the useful comments and suggestions that have helped us to improve our manuscript considerably. As indicated in the following responses, we have taken all your comments and suggestions into account in our revised manuscript. The line numbers appearing in the following responses indicate those of the text in the “Revised Manuscript with Track Changes” file.

Comments by reviewer #1.

Comment 1. 

Introduction, line 44:- replace “expecting” with “exciting” This line should read “…(CNP) analog is one of the most exciting therapeutic …”.

Response.

Thank you for your comment. We replaced from “expecting” to “exciting” in the revised version of our manuscript. Please confirm the revised sentence in line 44 in the Revised Manuscript with Track Changes.

Comment 2.

The reviewer suggest more background information regarding the rational for using a CNP/ghrelin chimeric peptide should be provided eg describe the potential for interaction with hyaluronic acid.

Response.

Thank you for your valuable suggestion. We added the following sentence about the background information of ASB20123 in line 55-60 of the Revised Manuscript with Track Changes;

 “The application of C-terminal part of ghrelin resulted in the higher stability of CNP analogs, compared to that of CNP-22 as the native form; it also improved their bioactivity as stimulators of endochondral bone growth. Furthermore, the C-terminal part of the ghrelin includes a BX7B motif, the cluster of basic amino acids might be implicated in the interaction with hyaluronic acid molecules which are the components of extracellular matrix in growth plate.”

In addition, we added 4 references in this part and it is shown in line 394-406 in the Revised Manuscript with Track Changes. 

Comment 3.

Page 12, line 191

There is no mention in the results section of any differences in blood chemistry measurements observed between the groups eg triglycerides (mentioned in the methods section). If there were no significant differences, the reviewer suggests line 191 could read “… ophthalmology, blood chemistry, urinalysis …”.

Response.

Thank you for pointing out. We missed describing the change of CPK activity. We added in line 196, 199-200 and 201 of the result section, in line 322-324 of the discussion section in the Revised Manuscript with Track Changes.

Comment 4.

It would be helpful for the reader if the age of the rats was included in the title of Supporting Table 1 or alternatively the title to the table read “Study 1 – histopathology in rats treated subcutaneously with …”.

Response.

Thank you for your comment. We added the study No. in the title of all figures and supporting table 1.

We are grateful to the Reviewer #2 for the critical comments and useful suggestions that have helped us to improve our manuscript considerably. As indicated in the following responses, we have taken all your comments and suggestions into account in our revised manuscript. The line numbers appearing in the following responses indicate those of the text in the “Revised Manuscript with Track Changes” file.

Comments by reviewer #2.

Major Comment.

Figure 4 of this manuscript shows a very interesting inverse relationship in response to ASB20123 infusion with the drug increasing bone length and decreasing density in both the cortical and trabecular bone. If this is the case, then it may limit the use of the drug because while it increases long bone length, it also results in less bone mineral density, which is unfavorable.

It would be very interesting to see if the BMD increases with time after the initial CNP analog infusion period has ended. In other words, if ASB20123 increases bone length, does it always also decrease bone mineral density or does it return to normal values after the ASB20123 treatment is discontinued. To test this, the investigators should analyze another group of animals 6 weeks after the last ASB20123 injection and determine if BMD is still reduced or whether it increased in the absence of the drug. The question being addressed with this experiment is whether any increase in bone length resulting from ASB20123 infusion also results in permanent reductions in BMD? If the later scenario is observed, then this could limit the utility of ASB20123 in the treatment of achondroplasia if it always results in less dense, and presumably, weaker bone.

Response.

Thank you for your valuable suggestion. It is very important to clarify the relationship the reduction of BMD and elongation of bone length caused by ASB20123 treatment. It is especially needed to care of the decrease of BMD in the treatment of achondroplasia, because the lower BMD in achondroplasia population was reported (Sims D, et al. Whole-body and segmental analysis of body composition in adult males with achondroplasia using dual X-ray absorptiometry. PLos One, 14(3): e0213806, 2019).

In this study, thickness of epiphyseal plate of femur, BMD and bone related toxicity were observed in the rats treated with ASB20123 at dose of 0.5 mg/kg/day and more for 4-week. Meanwhile, in a previous study, the body length increased significantly after continuous sc infusion at dose of 0.05 mg/kg/day and more for 12-week (Morozumi N, et al. ASB20123: A novel C-type natriuretic peptide derivatives for the treatment of growth failure and dwarfism. PLos One, 14(2): e0212680, 2019). At the same time, the BMD decreased significantly at dose of 0.15 mg/kg/day and more (internal data). We consider that these phenomena were caused mainly by rapid and excessive induction of bone growth in normal juvenile animals.

In addition, we evaluated the recovery of BMD and bone toxicity after 13-week recovery period. These results described in line 256-271 of result section and line 346-358 of discussion section in the revised version of our manuscript and added Fig. 5. The decreased BMD values induced by ASB20123 treatment did not return to the value of control group after 13-week recovery period. However, the BMD values in the ASB20123 treated groups increased similarly to those in the control group for 13 weeks. These results indicate that at least further reduction in BMD could be prevented by discontinuation of treatment.

Considering these results, the dosage regimen of CNP derivative would be a very important key factor for the success as a therapeutic drug.

Minor Comment.

The manuscript would benefit by editing from a native English-speaking person. Trabecular bone is the newer and more common term for “spongy” bone.

Response.

Thank you for pointing out. We replaced from “spongy” to “trabecular” in the revised version of our manuscript. Also, a native English-speaking person in our company have checked our revised manuscript.

---

## [Editor Report · Decision Letter 1]

27 Aug 2019

Safety assessment of a novel C-type natriuretic peptide derivative and the mechanism of bone- and cartilage-specific toxicity

PONE-D-19-14712R1

Dear Dr. Yotsumoto,

We are pleased to inform you that your manuscript has been judged scientifically suitable for publication and will be formally accepted for publication once it complies with all outstanding technical requirements.

With kind regards,

Michael Bader

Academic Editor

PLOS ONE
---

## [Editor Report · Acceptance letter]

3 Sep 2019

PONE-D-19-14712R1 

Safety assessment of a novel C-type natriuretic peptide derivative and the mechanism of bone- and cartilage-specific toxicity 

Dear Dr. Yotsumoto:

I am pleased to inform you that your manuscript has been deemed suitable for publication in PLOS ONE. Congratulations! Your manuscript is now with our production department. 

With kind regards,

on behalf of

Prof. Michael Bader 

Academic Editor

PLOS ONE